# Numb Chin Syndrome in Sickle Cell Disease: A Systematic Review and Recommendations for Investigation and Management

**DOI:** 10.3390/diagnostics12122933

**Published:** 2022-11-24

**Authors:** Mahdi Bedrouni, Lahoud Touma, Caroline Sauvé, Stephan Botez, Denis Soulières, Stéphanie Forté

**Affiliations:** 1Department of Physiology, McGill University, Montréal, QC H3A 0G4, Canada; 2Department of Neurosciences, Université de Montréal, Montréal, QC H3T 1J4, Canada; 3Library, Centre Hospitalier de l’Université de Montréal, Montréal, QC H2X 3E4, Canada; 4Departement of Medicine, Division of Hematology and Medical Oncology, Centre Hospitalier de l’Université de Montréal, Montréal, QC H2X 3E4, Canada; 5Department of Medicine, Université de Montréal, Montreal, QC H3T 1J4, Canada

**Keywords:** numb chin syndrome, sickle cell disease, sickle cell anemia, neuropathy, mental nerve, inferior alveolar nerve, paresthesia, hypesthesia, mandibular nerve

## Abstract

Numb chin syndrome (NCS) is a rare sensory neuropathy resulting from inferior alveolar or mental nerve injury. It manifests as hypoesthesia, paraesthesia, or, rarely, as pain in the chin and lower lip. Several case reports suggest that sickle cell disease (SCD) could be a cause of NCS. However, information about NCS is scarce in this population. Our objectives were to synthesize all the available literature relevant to NCS in SCD and to propose recommendations for diagnosis and management based on the best available evidence. A systematic review was performed on several databases to identify all relevant publications on NCS in adults and children with SCD. We identified 73 publications; fourteen reports met the inclusion/exclusion criteria. These described 33 unique patients. Most episodes of NCS occurred in the context of typical veno-occlusive crises that involved the mandibular area. Radiological signs of bone infarction were found on some imaging, but not all. Neuropathy management was mostly directed toward the underlying cause. Overall, these observations suggest that vaso-occlusion and bone infarction could be important pathophysiological mechanisms of NCS. However, depending on the individual context, we recommend a careful evaluation to rule out differential causes, including infections, local tumors, metastatic disease, and stroke.

## 1. Introduction

Sickle cell disease (SCD), the most common hemoglobinopathy worldwide, is a group of inherited blood cell disorders associated with episodes of acute illness and progressive organ damage [1]. Early historical data suggests that the prevalence of the sickle cell allele responsible for sickle cell disease is highest in Sub-Saharan Africa as well as some parts of the Mediterranean, the Middle East, and India reaching frequencies of up to 34%. Genetic studies and the strong geographic relationship between the distribution of the sickle cell allele and malaria suggest that the sickle cell allele confers a selective advantage for improving the survival of mutation carriers against severe and lethal malaria. Due to the modern globalization process, migration, and the forced relocation of millions of Africans to the Americas driven by the slave trade, means that the distribution of the sickle cell allele has expanded across the globe. Thus SCD has become a worldwide concern [2].

SCD has been associated with several central nervous system disorders, mainly through vascular phenomena [3,4]. Patients are at risk of silent ischemic infarcts, Moya Moya syndrome, aneurysms, and ischemic and/or hemorrhagic strokes [5,6,7,8]. An increased prevalence in the early onset of the neurocognitive disorder is observed, even in those without ischemic lesions, possibly because of reduced cerebrovascular reserve [9,10]. Patients with the SC genotype also commonly report auditory disturbances [11]. Despite peripheral neuropathies being relatively rare in SCD, there have been several cases associated with numb chin syndrome (NCS).

NCS is a rare sensory neuropathy characterized by inferior alveolar or mental nerve damage, which manifests as hypoesthesia, paraesthesia, or pain in the chin and lower lip (Figure 1) [12,13]. These nerves provide sensory information from the front of the chin, the lower lip, and the labial gingivae of the mandibular anterior teeth and the premolar teeth [14,15]. Although more commonly associated with malignancies, traumatic injuries, and dental interventions, NCS has also been described in patients suffering from SCD [16]. The few available case reports suggest that NCS was caused by SCD a complication of acute painful crises in the mandibular bone and infarction in the nerve passage within the mandibular canal. During an episode of vaso-occlusive crisis (VOC), it has been hypothesized that NCS can be triggered by local ischemia due to thrombosis of the vasa vasorum.

Given the limited published evidence on NCS in SCD, it remains a poorly characterized disease. Thus, information about the causes, disease course, and treatment is limited. Our objective was to review the available literature to synthesize all relevant studies that discuss NCS in SCD patients. A secondary objective was to propose recommendations for the management of NCS based on the best available evidence.

## 2. Materials and Methods

A systematic review of the currently available literature was performed following our pre-specified protocol that was registered on PROSPERO (CRD42021239583).

A number of databases were searched by a professional librarian for relevant studies on 5 January, 2021. The databases include MEDLINE (via Ovid, 1946 to 5 January 2021); Embase (via Ovid, 1974 to 3 January 2021), EBM Reviews (via Ovid up to 5 January 2021) and CI-NAHL Complete (via EBSCO, 1937 to 5 January 2021). The search strategies, designed by a librarian (CS), used text words and relevant indexing terms to identify studies concerning NCS in patients with SCD. The reports describing adults and/or children with any SCD genotype were included. Reports pertaining to patients with SCD traits were excluded. The MEDLINE strategy Appendix A was applied to all databases, with modifications to the search terms as necessary. No language limits were applied to the search. Conference abstracts of the relevant scientific meetings were manually searched with no language limits applied. Reviews, notes, editorials, or comments were excluded from the search criteria. The results were uploaded to Covidence, and duplicates were removed automatically. A two-stage screening process was performed to extract all the existing relevant reports. In the first step, two independent reviewers, (LT and MB) performed a title and abstract screening to identify, select, and filter the pertinent reports. In the second screening step, the two independent reviewers, (LT and MB) performed a full-text examination to further identify and select the pertinent reports. Excluded reports were sorted and designated as being “Non-relevant”, having the “Wrong study design”, or discussing an “Animal” study. A cross-comparison of the autonomously screened texts was carried out by the reviewers LT and MB. Disagreements were resolved by a consensus or with the input of a third reviewer (SF).

The extraction of the data from the final approved list of relevant texts was performed by the two independent reviewers (LT and MB). Discrepancies occurring within the data extraction step were resolved by a consensus or with the input of a third reviewer (SF). The study data that was extracted included the year of publication, country, study design, sample size, and follow-up duration. Patient demographics (including age, sex, country, and sickle cell type), clinical presentation, investigations, and outcomes were also extracted.

## 3. Results

After the removal of duplicates, 73 studies were identified using the pre-specified systematic search strategy. Following two levels of screening, 11 studies were included for extraction, and three publications were found and extracted following a manual search of references (Figure 2) [17,18,19,20,21,22,23,24,25,26,27,28,29,30]. The included studies were published between 1972 and 2021. Overall, 33 distinct patients were included in this report (Appendix A). The geographical distribution of patients was as follows: 39.4% of patients were from Jamaica, 15.2% from Ghana, 21.2% from the United States, 9.1% from France, 6.1% from India, 6.1% from England, and 3% from Turkey.

NCS due to SCD was reported across a wide range of ages (from 11 to 60 years), including five patients under the age of 18. The mean age at presentation was 28.3 years (SD = 11.7) (Table 1). Women represented 48.5% (16/33) of the patients. Sickle cell genotype distribution was reported as follows: 39.4% (13/33) had the hemoglobin SS sickle cell disease genotype, 15.2% (5/33) had the hemoglobin SC sickle cell disease genotype, 9.1% (3/33) had hemoglobin Sickle beta-(Sβ)-thalassemia, and the genotype was not reported for 36.4% of the patients. Hematocrit data was reported for six patients, with values ranging between 22.5% and 38.6% and a mean hematocrit of 30.4% (SD = 7.3).

Patients presented with either unilateral or bilateral chin numbness (Appendix A). The symptom distribution was as follows: 11 patients reported left-sided numbness or pain of the jaw or lip, 7 patients reported numbness on the right side, 10 patients presented with bilateral NCS, and symptoms were unclear for 5 patients. One patient reported the evolution of symptoms from the right side to both sides of the lower face and was included in the bilateral count. The duration of the neurological symptoms of 21 patients was reported in the literature, with a slight majority reporting the complete resolution of symptoms (52%, *n* = 11). The duration of symptoms ranged from 1.5 h to more than 168 months.

A number of acute medical conditions were occurring concomitantly or in the days prior to the NCS. Indeed, all but one case of NCS was associated with a vaso-occlusive crisis (VOC) and/or acute chest syndrome (ACS) (97%). In the case of VOC, the pain was described in the mandibular region in most cases (83%, *n* = 24). However, in five reports (27%), the pain was described in other regions, such as the hips, lumbar region, or knee, without involving the mandibular region. One case occurred post-dental surgery, which led to a VOC. One case was associated with osteomyelitis complicated by bone necrosis, and four others were associated with systemic infections. One patient presented in the context of multi-organ failure. In addition, some patients presenting with NCS had significant comorbidities. These included rheumatoid arthritis treated with prednisone (*n* = 1), pregnancy (*n* = 1), type II diabetes (*n* = 1), metastatic breast cancer treated by chemotherapy (*n* = 1), membranoproliferative glomerulonephritis (*n* = 1), asthma (*n* = 1), retinal detachment (*n* = 1), and splenectomy (*n* = 1).

Patients with NCS were investigated using a number of different imaging modalities (Appendix A). A specific clinical workup was reported for 16 patients. A mandibular X-ray was used in 10 patients, making it the most used imaging modality. Five of these patients presented with specific findings, including (1) focal radiolucency, (2) diffuse lytic changes, and (3) stepladder trabeculations, consistent with (1) bone infarction, (2) osteomyelitis and (3) reactive bone changes typical of sickle cell disease. Images for the remaining five patients displayed no abnormalities. Seven patients were examined using a head CT scan. In cases where a patient was tested with both a mandibular X-ray and a head CT scan, the head CT scan did not reveal any notable difference or offer any alternative explanations for the mental nerve neuropathy. In addition, seven patients were examined with an MRI (one MRI exam was self-reported by a patient who claimed to have been previously assessed; the data was unavailable). Abnormalities were reported in 2 of these patients, including one incidental brain lesion unrelated to the NCS. Furthermore, a radionuclide bone scan was performed on 3 patients, which positively detected abnormalities in 2 patients. One patient had slightly increased tracer uptake in the right mandibular molar region, indicative of bone infarction. The other patient had diffusely increased tracer uptake in the skull and the periarticular regions of the long bones, consistent with reactive bone marrow hyperplasia. A lumbar puncture was performed on 1 patient showing no abnormalities suggestive of infection or malignancy. Finally, a dental pulp test was performed on 1 patient, which marked the non-vitality of several teeth. 

The treatments were mainly symptomatic, with the management of the underlying cause of the neuropathy mainly being VOC/ACS. Treatments were reported for 18 patients. These included blood transfusions in two patients, tooth extractions in one patient, antibiotics in the case of infections in two patients, and standard treatment of VOC, including hydration, oxygen administration, and analgesics in 15 patients. Of note, one patient required no treatment.

## 4. Discussion

Our systematic review presents the largest collection of patients with NCS secondary to SCD. NCS seems to affect patients of all ages, sex, and SCD genotype. The majority of events occurred in the context of VOC/ACS, highlighting the important relationship between SCD and NCS pathophysiology.

### 4.1. Pathophysiology

Some pathophysiological mechanisms may be common to both VOC and NCS. Indeed, in an early report on the topic of this mental-nerve neuropathy, it was noted that the mandible was the site of merely 4% of the 100 consecutive adult admissions for a painful sickle-cell crisis [17]. However, in a more recent case series, 13 patients were presented with a number of concurrent diagnoses, such as active infection, rheumatoid arthritis, and diabetes, that could have either (1) contributed to the development of the VOC which in turn caused the NCS, or (2) these could be independent causes of NCS regardless of VOC [27]. This highlights the importance of careful clinical evaluation to identify both causes of VOC and NCS.

Vaso-occlusion and the resulting tissue ischemia are unique features of SCD resulting from the presence of the pathological hemoglobin S. A point mutation in the beta-globin gene on chromosome 11 leads to the replacement of a glutamic acid residue with a valine residue on the surface of the protein. The resulting hemoglobin is characterized by abnormal hydrophobic interactions with the adjacent chains, leading to polymer bundling and the distorted red blood cell shape distinctive of the sickle cells. The passage of these crescent-shaped red blood cells through the narrow blood vessels is impaired, which may lead to ischemia and hemolysis [31]. Common triggers of vaso-occlusion are inflammatory or infectious conditions, such as infection, hypoxia, dehydration, and acidosis [1].

The anatomical location of the vaso-occlusion in the case of NCS could be the mandibular bone. The inferior alveolar nerve that branches into the mental nerve and the inferior alveolar artery, and the main blood supply to the mandible bone, are characterized by their passage into the mandibular foramen, through a narrow mandibular bony canal, before exiting at the mental foramen. The nature of this confined anatomical structure makes it susceptible to compression, ischemic infarction, and infection. Sickle red blood cells could block the small inferior alveolar artery and prevent perfusion into the inferior alveolar and mental nerves [14,23,24,27]. Additionally, edema due to bone infarction and osteomyelitis could compress these structures. Numbness would thus typically occur in the area of the chin and lower lip, where the mental nerve, the terminal branch of the alveolar nerve, provides sensory innervation [14]. Some studies described in this review have documented imaging consistent with bone infarction in this anatomical location [19,26].

The vasa nervorum is another potential locus of vascular injury in the case of NCS associated with SCD. SCD leads to a generalized inflammatory state and widespread vascular changes that can, in turn, result in vascular complications typical to both large (e.g., stroke) and small (e.g., retinopathy) blood vessels. In other medical conditions, such as diabetes or connective tissue disease, vasculitis in these small blood vessels compromise the vascular supply and can lead to neuropathy involving single nerves [32,33].

Theoretically, it is conceivable that hyperviscosity could also contribute to NCS pathophysiology [34]. In SC disease, a subtype of sickle cell disease was characterized by more elevated hemoglobin and a more silent and insidious disease course; it has been proposed that phlebotomies and the resulting anemia could prevent the occurrence or progression of neurosensory hearing loss and other complications attributed to hyperviscosity [35,36].

In addition, SCD patients are significantly more susceptible to infections, severe infectious complications, and osteomyelitis [37]. Serious bacterial infections remain a major cause of death in low- and medium-income countries [38]. Indeed, it is postulated that early and progressive injury to the spleen in SCD patients as a consequence of repeated vaso-occlusion followed by ischemia leads to progressive fibrosis and atrophy of the spleen [39]. For instance, one patient affected by local osteomyelitis led to NCS [21]. The resultant swelling and necrosis may have compressed and damaged the distribution of the left inferior alveolar and mental nerves resulting in mandibular neuropathy. Thus, a combination of VOC and a vulnerability to infection and osteomyelitis may predispose SCD patients to NCS.

In the general population, NCS has been associated with metastatic disorders. In our review, there was only one reported case of a patient developing NCS in the context of metastatic disease to the liver secondary to breast cancer. The associated pathophysiological mechanism is different than in the case of VOC, as neuropathy is caused by the compression or infiltration of the mental or inferior alveolar nerves by neoplastic cells [18]. In this reported case, NCS may be symptomatic of perineural invasion relating to metastatic liver cancer.

### 4.2. Differential Diagnosis

Before concluding that a case of NCS is caused by SCD, a number of alternative diagnoses that can cause NCS in the general population must be rigorously considered. These diagnoses have been comprehensively reviewed elsewhere [16]. In particular, dental procedures and local trauma are the most common cause of NCS and are usually obvious in medical history. Moreover, based on anatomy, the differential diagnosis must include odontogenic infection, osteomyelitis, osteosarcoma, primary lymphoma of the bone, and central mucoepidermoid carcinoma. In the absence of clear anatomic etiology, infectious, autoimmune, inflammatory, toxic, and malignant etiologies should be considered. Epidemiological risk factors and clinical suspicion will guide the relevant work-up. Of interest, sarcoidosis has been reported as one of the common causes of NCD. Patients of African descent are at increased risk of developing sarcoidosis [40].

In addition, systemic infections can trigger VOC/ACS and must be considered in patients presenting with NCS. Given the association of NCS with metastasis, systemic neoplasms should be considered in all patients, although a neoplasm was noted in only one case in this review. SCD patients are at risk of ischemic and hemorrhagic strokes, but vascular events would unlikely present with only sensory symptoms in the mental nerve distribution. Patients are also at risk of electrolyte disorders, namely hypocalcaemia, secondary to blood transfusions which can present with circumoral numbness or paresthesia. These are typically not lateralized or as delineated as NCS and resolve with the correction of the hypocalcaemia.

### 4.3. Imaging

The most frequently used imaging modalities in these studies were mandibular X-rays and head CT scans, followed by cerebral MRI and bone scans. One patient who was examined using a CT scan and whose results appeared normal was also assessed using a cerebral MRI that displayed subperiosteal fluid collection in the mandibular rami. The imaging by CT scan offers a higher level of detail than X-rays while still maintaining high imaging speeds and is generally accessible. However, in this review, data from patients tested using both imaging modalities showed no significant difference in the findings. CT scans would still be recommended over X-ray imaging, if available, for a more sensitive and accurate diagnosis of bone destruction and masses [16]. In turn, cerebral MRIs are tools that offer more detailed imaging of brain parenchyma and the pathway of the trigeminal nerve compared to CT scans [41,42]. Importantly, the mandibular region must be included in the scanned region, which is not always the case with a standard brain MRI; otherwise, important anatomical findings may be missed [43]. Although central neurological causes of the numb chin are probably extremely rare, a thalamic lacunar infarct can present such circumscribed symptoms [44]. Since patients with sickle cell disease have a heightened risk of stroke, this possibility should be considered. In our study, MRIs were performed only in the United States, France, England, and India. This suggests that the geographical location may have influenced access to investigations. MRIs may be less accessible in regions where SCD is most prevalent [45]. In these contexts, other imaging modalities should be used, including mandibular X-ray and/or CT scans.

### 4.4. Treatment

Currently, there is no specific treatment for NCS, and the literature is even scarcer with respect to cases with SCD. Indeed, the management of sensory neuropathy was directed toward resolving the underlying cause. Patients typically received the standard treatment for VOC to resolve their crisis, including hydration and analgesia. Some patients receive blood transfusions. However, this is not a standard treatment for uncomplicated VOC. Furthermore, the role of transfusions in the treatment of NCS is unknown. The patient who received a tooth extraction suffered from a VOC and infection in the area between the first and third molar, which were promptly treated. In this respect, comorbidities, such as infections or malignancies, must be treated parallel to the treatment of the VOC. Indeed, several series highlights the importance of assessing concurrent illnesses within SCD patients who present with NCS, as it can be the first manifestation of systemic disorders [27].

Finally, in the case of neuropathic pain, Pregabalin or Gabapentin could be tried in analogy to herpes zoster reactivation for pain relief [46]. Other therapeutic options include tricyclic antidepressants (ex: Amitriptyline, Nortriptyline), other antiepileptics (ex: Lamotrigine), and neural blockade. However, no data is available on the effectiveness of these medications in the case of NCS related to SCD.

### 4.5. Recommendations

Based on the limited available evidence and our collective clinical expertise, we recommend a thorough evaluation of patients presenting with NCS in SCD by a neurologist and a clinician with expertise in SCD (hematology, internal medicine, pediatrician) (Table 2). We consider a detailed oral cavity examination by a dentist or ENT essential, especially in the case of a history of recent dental procedures, trauma, or pain. A detailed medical history should also be assessed for symptoms of systemic infection, potential malignancy, or auto-immune disease, along with common VOC/ACS precipitants. In addition to a general physical examination, a complete neurological exam is necessary to delineate the sensory deficit and rule-out other focal deficits. For all patients, laboratory studies should minimally include a complete blood count to assess the severity of the anemia, biochemistry to screen for kidney failure, and a pregnancy test since pregnancy can be a trigger of VOC and other acute complications of sickle cell disease. Furthermore, certain antibiotics that are commonly used to treat SCD complications are to be avoided in pregnancy (e.g., tetracyclines and fluoroquinolones). Therefore, according to the physicians’ judgment, a pregnancy test may be ordered. In the case of fever, chest pain, or any respiratory sign/symptom, an urgent X-Ray should be performed to identify acute chest syndrome. This is a potentially deadly medical emergency mandating specific care not covered in this article but described extensively elsewhere [47].

Further testing that can be consider in cases of atypical presentations are highlighted in Table 2.

## 5. Conclusions

This study represents the largest review to date of the clinical manifestations, pathophysiology, investigations, and management of NCS in patients with SCD. NCS occurred most often in the context of VOC/ACS. The data extracted from this systematic review reinforces the hypothesis that a vaso-occlusion affecting the inferior alveolar or mental nerves in the microvasculature traversing through the mandibular bone could be a pathophysiological mechanism of NCS. In most cases, the management consisted of treating the acute trigger of the NCS and providing adequate analgesia. Careful evaluation is warranted to rule out alternative differential causes of NCS, including local infection, primary neoplasm, or metastatic disease. Imaging serves to narrow down the differential diagnoses and identify treatable causes. This study brings attention to a rare and underrecognized complication of SCD that deserves further investigation to optimize diagnosis and management.

## Figures and Tables

**Figure 1 diagnostics-12-02933-f001:**
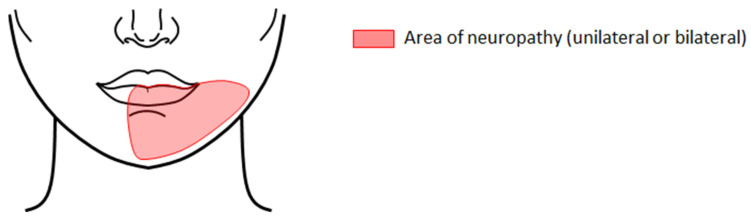
Area of sensory neuropathy in numb chin syndrome (adapted from Konotey-Ahulu, 1972 [17]).

**Figure 2 diagnostics-12-02933-f002:**
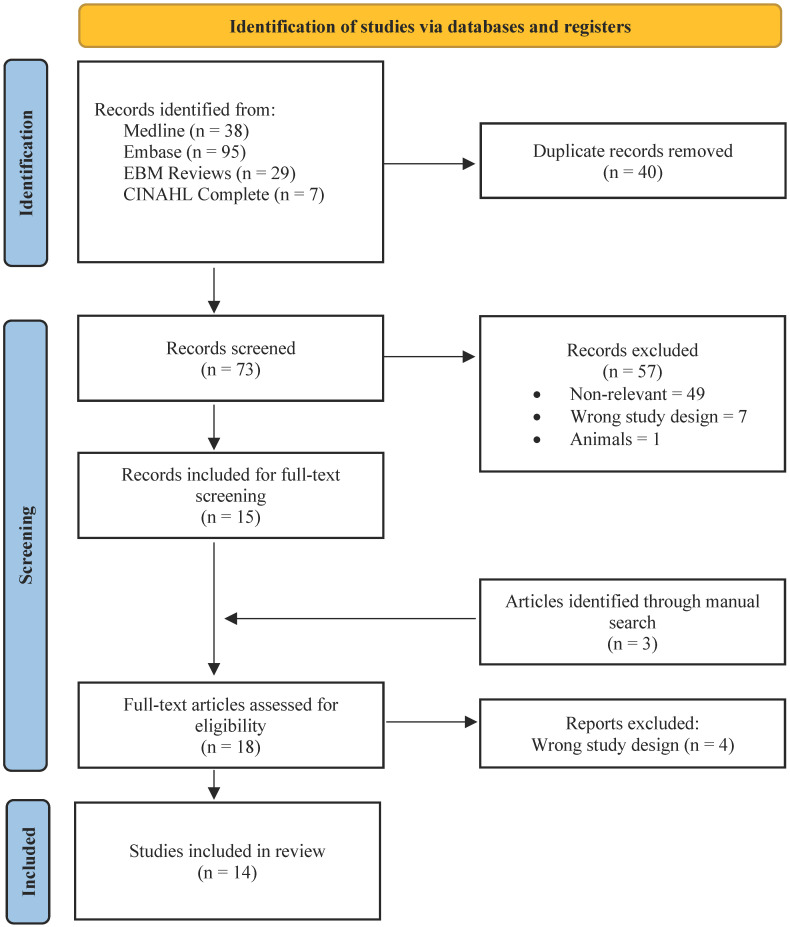
PRISMA flow diagram.

**Table 1 diagnostics-12-02933-t001:** Patient characteristics, NCS presentation, and management.

Patient Characteristics (*n* = 33)		
Age, years (mean) (SD)		28.3 (11.7)
Sex, *n* (%)	Male	17 (51.5)
	Female	16 (48.5)
Genotype, *n* (%)	SS	13 (39.4) *
	SC	5 (15.2) *
	Sβ-thalassemia	3 (9.1) *
	Data unavailable	12 (36.4) *
Comorbidities ^+^, *n* (%)	None	27 (82)
	Diabetes	1 (3)
	Rheumatoid arthritis	1 (3)
	Asthma	1 (3)
	Membranoproliferative glomerulonephritis	1 (3)
	Splenectomy	1 (3)
	Retinal detachment	1 (3)
	Malignancy	1 (3)
Numb chin presentation and management		
Location, *n* (%)	Unilateral	18 (54.5)
	Bilateral	10 (30.3)
	Unclear	5 (15.2)
Resolution of symptoms, *n* (%)	Yes	11 (33.3)
	No	10 (30.3)
	Data unavailable	12 (36.4)
Acute medical condition ^+^, *n* (%)	None	1 (3)
	Veno-occlusive crisis/Acute chest syndrome	32 (97)
	Osteomyelitis	1 (3)
	Infection	5 (15)
	Multiorgan failure	1 (3)
	Osteonecrosis of the mandible	1 (3)
Treatment ^+^, *n* (%)	Standard treatment of VOC	15 (45)
	Treatment for infection	2 (6)
	Transfusion	2 (6)
	No treatment	1 (3)
	Data unavailable	14 (42)
Geographical distribution, *n* (%)	Jamaica	13 (39.4)
	United States of America	7 (21.2)
	Ghana	5 (15.2)
	France	3 (9.1)
	India	2 (6.1)
	England	2 (6.1)
	Turkey	1 (3)
Duration		Min: 0.03 months Max: no resolution after 14 years

^+^ Patients can have more than one comorbidity, acute medical condition, or treatment; * Do not add up to 100% due to rounding; Abbreviations: Sβ-thalassemia (hemoglobin sickle-beta thalassemia), SC (hemoglobin SC sickle cell disease), SS (hemoglobin SS sickle cell disease).

**Table 2 diagnostics-12-02933-t002:** Proposed work-up for numb chin syndrome in sickle cell disease.

Minimal Work-Up	
Clinical	Detailed history (trauma, infection, malignancy, autoimmune disease, VOC) General physical exam Detailed neurological exam Oral cavity examination by dentist or ENT
Biological	Complete blood count Creatinine Pregnancy test
Imaging	Xray of mandible or CT scan
**Further diagnostic tests to consider**	
Biological	Diabetes screening (Blood glucose or fructosamine/protein ratio) * Plasma protein electrophoresis with immunofixation, free light chains, cryoglobulin Serological tests for auto-immune diseases (ANA, ENA, Ro, La) Syphilis, HIV, HSV, Lyme testing Lumbar puncture (protein, cytology, flow cytometry)
Imaging	CT scan MRI PET scan

Abbreviations: ENT = Ear nose throat specialist; MRI = magnetic resonance imaging; VOC = Veno-occlusive crisis. * standard diabetes screening using Hb1Ac is not reliable in patients with hemoglobinopathies.

## Data Availability

Not applicable.

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
