# Peer review of "Numb Chin Syndrome in Sickle Cell Disease: A Systematic Review and Recommendations for Investigation and Management"

_diagnostics, 2022, doi:10.3390/diagnostics12122933_

Round 1

Reviewer 1 Report

Well presented comprehensive review

Author Response

Dear Reviewer,

Thank you

Reviewer 2 Report

This systematic review describes the rare and likely under-recognized condition of numb chin syndrome (NCS) in the context of Sickle Cell Disease (SCD). Given the rarity of this condition the compilation of data is noteworthy, however a few comments need to be addressed to improve suitability for publication

1) Page 4: 'Sickle cell genotype distribution was reported as follows: 39% (13/33) had the SS genotype, 112 15% (5/33) had the SC genotype, 9% (3/33) Sβ-thalassemia, and genotype was not reported 113 for 12% of patients.' This does not add up to 100%!

2) Table 1: Several of the figures in brackets do not add up to 100%, including Treatment (total is 102%). Also the 'acute medical condition' and 'comorbidities' similarly do not add up to 100%, however this may be due to some patients having more than 1 comorbidity or more than 1 acute medical condition. I suggest that, given the small size of this cohort the number of patients with >1 comorbidity/medical condition is delineated separately for clarity. 

3) a)Table 1/elsewhere: the geographical location of the case reports included should be documented in the manuscript/table e.g. USA/Canada/Europe/Africa.  

b) Similarly, later when the NCS investigations are being described, the authors should document whether these differed based on the geographical location of the patient and whether socioeconomic environment may have influenced the access to investigations.

4) The authors describe age range from 11-60 years. However they should additionally divide this into the total number of paediatric versus adult patients for clarity. Furthermore, the authors should comment on whether there were any differences between paediatric and adult patients in terms of presentation, management or duration of NCS?

5) The authors say that (one MRI was self-reported). Can they please explain what this means? Was the patient a radiologist or was the source of the case report directly obtained from a patient, who may or may not have fully understood the results obtained?

6) Page 6 Discusssion: 'In our cohort, NCS seems to affect patients of all sickle cell genotypes and with varying 212 hematocrit levels, therefore hyperviscosity may not be a major driver of neural injury in 213 the case of NCS in SCD.'

This is too bold a statement. Given that there are only 33 patients in total, of whom 5 are HbSC, the reviewed cohort is far too small to point to any underlying pathophysiology. Please remove this statement

7) Page 8 discussion: Although recommending MRI for investigation, the authors also acknowledge 'that MRIs may not be accessible in regions where SCD is most prevalent [45].'

Therefore a second option should be suggested as the preferred imaging modality if MRI access is not possible.

8) Can the authors describe the rationale for a pregnancy test in the context of NCS?

Author Response

Dear Reviewer,

Thank you

Reviewer 3 Report

In the introduction section the authors should point out that NCS is a rare disease. This would justify the low number of patients analyzed.

In addition to bacterial infections, are these patients prone to viral infections (eg HERPES)?

If they are subject, it should be said in the text.

The first time that an abbreviation appears in the text, it is preferable to write the name in full (e.g., genotype SS, line 112).

Add abbreviations below table 1.

A list of abbreviations is required. 

Considering that SCD is associated with inflammation and that CRP is an inflammatory marker, the evaluation of this marker should be added to table3. (Mohammed FA, Mahdi N, Sater MA, Al-Ola K, Almawi WY. The relation of C-reactive protein to vasoocclusive crisis in children with sickle cell disease. Blood Cells Mol Dis. 2010 Dec 15;45(4):293-6. doi: 10.1016/j.bcmd.2010.08.003. Epub 2010 Sep 1. PMID: 20813565).

Author Response

Dear Reviewer,

Thank you

Reviewer 4 Report

Very well organised review, worth publising as it is

Author Response

Dear Reviewer,

Thank you
